# Discrepancy Analysis between Histology and Molecular Diagnoses in Kidney Allograft Biopsies: A Single-Center Experience

**DOI:** 10.3390/ijms241813817

**Published:** 2023-09-07

**Authors:** Liye Suo, Martha Caicedo Murillo, Brian Gallay, Reut Hod-Dvorai

**Affiliations:** 1Department of Pathology, The State University of New York Upstate Medical University (SUNY Upstate Medical University), Syracuse, NY 13210, USA; caicedom@upstate.edu; 2Division of Nephrology and Transplantation, The State University of New York Upstate Medical University (SUNY Upstate Medical University), Syracuse, NY 13210, USA; gallayb@upstate.edu

**Keywords:** kidney transplant, histology, infection, molecular diagnosis

## Abstract

Histology diagnosis is essential for the monitoring and management of kidney transplant patients. Nowadays, the accuracy and reproducibility of histology have been criticized when compared with molecular microscopy diagnostic system (MMDx). Our cohort included 95 renal allograft biopsies with both histology and molecular diagnoses. Discrepancies between histology and molecular diagnosis were assessed for each biopsy. Among the 95 kidney allograft biopsies, a total of 6 cases (6%) showed clear (*n* = 4) or borderline (*n* = 2) discrepancies between histology and molecular diagnoses. Four out of the six (67%) were cases with pathologically and clinically confirmed active infections that were diagnosed as mild to moderate T-cell-mediated rejection (TCMR) with MMDx. Two cases showed pathological changes that were not sufficient to make a definitive diagnosis of active rejection via histology, while MMDx results showed antibody-mediated rejection (ABMR). In addition, there were six cases with recurrent or de novo glomerular diseases diagnosed only via histology. All other biopsy results were in an agreement. Our results indicate that histology diagnosis of kidney allograft biopsy is superior to molecular diagnosis in the setting of infections and glomerular diseases; however, MMDx can provide helpful information to confirm the diagnosis of active ABMR.

## 1. Introduction

Histology diagnosis is essential for the monitoring and management of kidney transplant patients [1]. Although histopathology with the Banff classification system remains the gold standard approach for diagnosing T-cell-mediated rejection (TCMR is characterized by interstitial and epithelial infiltration of T-cells and macrophages in the allograft kidney) and antibody-mediated rejection (ABMR) in kidney allograft biopsies [2,3], there are limitations to this strategy: there are currently no external standards available for validation of the Banff criteria, and the scoring is considered semiquantitative with less accuracy and more inter and intra-observer variability [4]. Hence, the accuracy and reproducibility of histology have been criticized when compared with molecular diagnosis [5,6,7]. 

Molecular diagnosis, a potential alternative to histopathology, is believed to have better diagnostic accuracy and reproducibility in terms of rejection [5]. The molecular microscope diagnostic system (MMDx) measures the global gene expression related with rejection and injury in biopsy tissues. This technology has the advantage of being objective and can provide insight into the pathophysiology of diseases based on gene expression [5,8]. However, MMDx also has limitations in the diagnosis of kidney allograft biopsy. For example, MMDx cannot directly diagnose glomerular diseases, infections or post-transplant lymphoproliferative disorder (PTLD), which can be crucial for guiding patient clinical management. Currently, due to these limitations, MMDx cannot completely replace histology diagnosis for kidney allograft biopsies, and therefore, it is typically being utilized in tandem with histology. A significant issue pertaining to the identification of kidney transplant rejection and patient management is the discrepancy in findings between these two methods.

Previous studies investigating the discrepancies between histology and MMDx indicate discrepancy rates ranging from 20 to 35% [5,9]. These rates may be even higher when histopathology is ambiguous and more open to subjective interpretations. Similar discrepancies exist among renal pathologists applying the Banff criteria, with a sensitivity of one pathologist to another ranging from 45 to 56% [9]. One of the largest cohort studies in discrepancy analysis, designed by Alberta Transplant Applied Genomics, revealed that histology disagreed with MMDx in 37% of kidney allograft biopsies [5]; it also showed that molecular diagnosis demonstrates greater sensitivity and negative predictive value than histopathology with unambiguous diagnoses. In addition, although the authors indicated that the molecular changes between BK polyomavirus-associated nephropathy and TCMR can overlap [5,10,11,12], they emphasized that TCMR diagnosis from MMDx is clinically meaningful for guiding patient management and indicated that the TCMR may resolve without treatment when full immunosuppression is restored [5]. 

At our center, MMDx is routinely sent-out and performed at Kashi Clinical Laboratories when kidney allograft biopsies are indicated. The goal of this study was to characterize clinically significant discrepancies between histology and molecular diagnoses in kidney allograft biopsies in order to guide the management of our transplant recipients. 

## 2. Results

Among the 95 kidney allograft biopsies, a total of 6 cases (6%) showed clear (*n* = 4) or borderline (*n* = 2) discrepancies between histology and molecular diagnoses (Table 1). In addition, there were six cases with recurrent or de novo glomerular diseases diagnosed only via histology: two cases of thrombotic microangiopathy (TMA), one case of recurrent membranous glomerulopathy, one case of recurrent IgA nephropathy, one case of collapsing glomerulopathy and one case of infection-related glomerulonephritis. All other biopsy results were in an agreement. 

All four cases which exhibited clear discrepancies showed pathologically and clinically confirmed active infections. MMDx detected mild to moderate T-cell-mediated rejection (TCMR): two cases of active BK nephropathy (Figure 1); one case of active CMV nephropathy (Figure 2) and one case of active pyelonephritis (Figure 3). 

### 2.1. Case of Active BK Nephropathy

A 66-year-old man who received kidney transplant from a deceased donor presented with acute kidney injury with elevated serum creatinine of 4.86 mg/dL and positive serum BK quantitative PCR of 1,900,000 IU/mL. Histology showed focal interstitial inflammation with marked tubulitis and abundant nuclear viral inclusions in the proximal tubular epithelial cells which are strongly positive for SV40 immunohistochemical staining (Figure 1). These findings confirmed the diagnosis of active BK nephropathy. However, MMDx showed moderate TCMR and no ABMR, which was considered as false positive TCMR result. After the biopsy, aggressive management of BK nephropathy was initiated and the serum creatinine level decreased. 

### 2.2. Case of Active CMV Nephropathy

A 65-year-old man who received kidney transplant from a brain-dead donor presented with progressive dyspepsia, daily vomiting and failure to thrive. He also presented with acute kidney injury with elevated serum creatinine of 4.2 mg/dL. Histology showed focal glomerulitis with few podocytes, nuclear viral inclusions and positive CMV immunohistochemical staining. No interstitial inflammation or tubulitis was present (Figure 2). These findings confirmed the diagnosis of active CMV nephropathy. Later, the patient’s serum CMV quantitative PCR showed a positive result of 403,000 IU/mL. However, MMDx showed moderate TCMR and no ABMR, which was considered as false positive TCMR result. After the biopsy, aggressive management of antiviral therapy (valganciclovir) was initiated. CMV titer significantly decreased to <200 IU/mL and serum creatinine level was back to his baseline. 

### 2.3. Case of Active Pyelonephritis

A 62-year-old woman who received kidney transplant from a brain-dead donor presented with acute kidney injury with elevated serum creatinine of 4.6 mg/dL. Urine analysis showed abundant white blood cells with 3+ leukocyte esterase; urine culture showed greater than 100,000 col/mL *E. coli*. Histology revealed acute tubular epithelial injury with focal neutrophilic rimming along proximal tubules and microabscesses (Figure 3), which are consistent with active pyelonephritis. However, MMDx showed results of mild TCMR and no ABMR. Later, the patient was treated with antibiotics and the renal function was recovered after the treatment. 

Two cases in the group of borderline discrepancy showed pathologic changes which were not sufficient to make a definitive diagnosis of active rejection via histopathology (Figure 4), while MMDx results showed antibody-mediated rejection (ABMR) (Table 1).

### 2.4. Case of Isolated Endarteritis

A 51-year-old man who received kidney transplant from a brain-dead donor presented with acute kidney injury with elevated serum creatinine of 1.47 mg/dL. Donor-specific antibodies (DSAs) were negative. Histology revealed one artery with mild endarteritis in the background of normal-appearing renal cortex (Figure 4). There was no interstitial inflammation, no tubulitis, no glomerulitis, no peritubular capillaritis with negative C4d immunohistochemical staining. Concurrent MMDx showed the result of moderate early-stage ABMR and no TCMR. The patient was treated with methylprednisolone 500 mg intravenously for a total of 3 doses. Re-biopsy after two weeks exhibited no histologic features of active rejection or no endarteritis. MMDx affirmed the result of no ABMR and no TCMR. DSA remained as negative. 

### 2.5. Case of Mild Peritubular Capillarity with Negative C4d Staining

A 39-year-old man who received kidney transplant from a living-unrelated donor presented with higher than expected Allosure (dd-cfDNA of 2.3%, but with serum creatine level of 1.4 mg/dL. DSA (donor-specific anti-HLA antibodies) were positive. Histology revealed focal mild peritubular capillaritis with no glomerulitis and negative C4d immunohistochemical staining. No interstitial inflammation or tubulitis was present. These findings were not sufficient to make the diagnosis of ABMR based on the Banff criteria. However, MMDx later showed early mild ABMR. The patient underwent pheresis with IVIG for further diagnosis with ABMR. After 3 weeks, dd-cfDNA decreased to 0.9% and serum creatinine decreased to 1.1 mg/dL. 

## 3. Methods

We identified all renal transplant recipients on whom renal allograft biopsies were performed at our center between November 2021 and November 2022, who had concurrent testing for molecular microscopy diagnostic system (MMDx) (*n* = 95). A retrospective chart review was performed to obtain the pertinent clinical information, including clinical diagnosis, indication for allograft biopsy, serology testing for infectious diseases, urine culture, HLA antibody and the treatment after the diagnosis. 

The histology slides were retrieved and evaluated by the renal pathologist in our center; the MMDx testings were performed at Kashi Clinical Laboratories at Portland, Oregon. Discrepancies between histology and molecular diagnosis were assessed for each biopsy. 

Our study received the ethics approval (2023524-1) from the SUNY Upstate institutional review board.

## 4. Discussion

Our study highlights that MMDx diagnosis can produce false positive results in the setting of active infections, especially viral infections. The MMDx study group indicated that the diagnosis of TCMR via MMDx in the context of BK nephropathy can be clinically significant, and TCMR may resolve without treatment when full immunosuppression is restored [5]. Here, we demonstrated that in certain cases of active viral infections, including polyomavirus and CMV, transcripts detected via MMDx can be interpreted as TCMR; in these cases, the diagnosis of TCMR with MMDx can be misleading and result in over-immunosuppression of the patient. We acknowledge that histology cannot always distinguish between TCMR and viral infection, and that concurrent TCMR and infection may exist. However, the clinical treatment response and pathologic features in the cases presented here suggest that the TCMR diagnosed with MMDx was false positive (e.g., the active CMV case was only treated with anti-viral therapy not targeting the “TCMR”, and the patient recovered).

Prior reports and studies [5,8,9,13,14] emphasized the certainty and unambiguity of MMDx diagnosis while criticizing traditional histology diagnosis for lack of reproducibility and less agreement with clinical judgment. However, in our study, we identified six discrepant cases. Of those, in five (83%) cases (four with active infection and one with isolated endarteritis), the clinical team followed a management plan based on histopathology findings; only one case was treated based on MMDx diagnosis of early ABMR without sufficient histology findings. Our data suggest that “ambiguous” or descriptive findings from histopathology can be clinically critical to guide the treatment plan. 

On the other hand, MMDx has the advantages of early detection and unambiguous confirmation of ABMR diagnosis when histology findings are not sufficient to make the diagnosis. Individual histological features of acute ABMR, as described in the Banff classification, are not specific and sometimes are not sufficient for a definitive diagnosis of ABMR in the absence of other features such as the presence of DSA or C4d staining. In the updated Banff classification, validated gene transcript assays such as MMDx have been accepted in support of the diagnosis of ABMR. Understanding of the allograft rejection process is still evolving [15], especially for cases showing isolated microvascular inflammation on histology (negative DSA and C4d staining). Future studies analyzing data from both histology and gene transcript assays have the potential to help in defining the underlying mechanisms. 

Our study has several limitations. This is a single-center retrospective study with a limited sample size. Our study focused only on the discrepancies of the main diagnoses from both methods. We acknowledge that more detailed data from both methods, such as the evaluation for chronicity (interstitial fibrosis with tubular atrophy) and the severity of acute kidney injury, should be compared in the future. 

In conclusion, histopathology is still superior to molecular diagnosis in the setting of diagnosing active infections and glomerular diseases; on the other hand, MMDx can provide helpful information to confirm the diagnosis of early-active ABMR. The utilization of both methods is beneficial for increasing the accuracy and certainty of diagnosis in kidney allograft biopsies. 

## Figures and Tables

**Figure 1 ijms-24-13817-f001:**
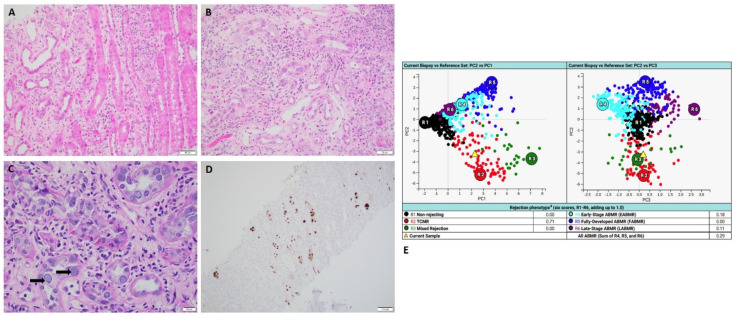
A case of active BK nephropathy with MMDx diagnosis of TCMR. (**A**,**B**) Hemoxylin and Eosin stains showed marked interstitial inflammation with severe tubulitis (scale bar is 50 µm). (**C**) PAS stain showed abundant viral nuclear inclusions (black arrows) (scale bar is 20 µm). (**D**) SV40 immunohistochemical staining showed abundant positive nuclear staining for BK polyomavirus (scale bar is 100 µm). (**E**) MMDx showed moderate TCMR. (MMDx: molecular microscopy diagnostic system; TCMR: T-cell-mediated rejection).

**Figure 2 ijms-24-13817-f002:**
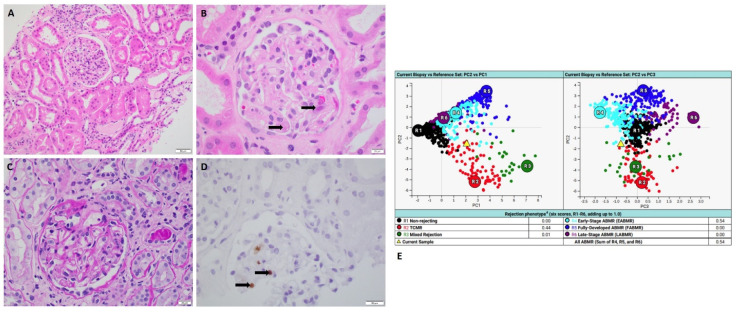
A case of active CMV nephropathy with MMDx diagnosis of TCMR. (**A**,**B**) Hemoxylin and Eosin stains showed glomerulitis with no tubulitis or interstitial inflammation (Black arrows highlight viral nuclear inclusions in podocytes) (scale bar for A is 50 µm; scale bar for B is 20 µm). (**C**) PAS stain showed glomerulitis (scale bar is 20 µm) (**D**) CMV immunohistochemical staining showed positive nuclear staining (black arrows) for CMV in the focal podocytes (scale bar is 20 µm). (**E**) MMDx showed moderate TCMR. (MMDx: molecular microscopy diagnostic system; TCMR: T-cell-mediated rejection).

**Figure 3 ijms-24-13817-f003:**
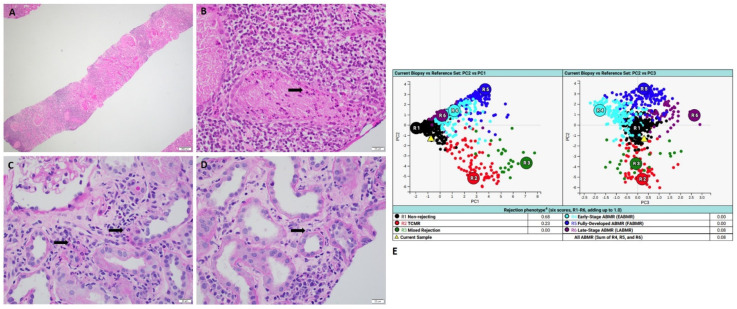
A case of active pyelonephritis with MMDx diagnosis of TCMR. (**A**,**B**) Hemoxylin and Eosin stains showed diffuse interstitial inflammation with neutrophilic microabscesses (black arrow) (scale bar for A is 200 µm; scale bar for B is 20 µm). (**C**,**D**) PAS stain showed interstitial inflammation with neutrophilic rimming (black arrows) (scale bar is 20 µm). (**E**) MMDx showed mild TCMR. (MMDx: molecular microscopy diagnostic system; TCMR: T-cell-mediated rejection).

**Figure 4 ijms-24-13817-f004:**
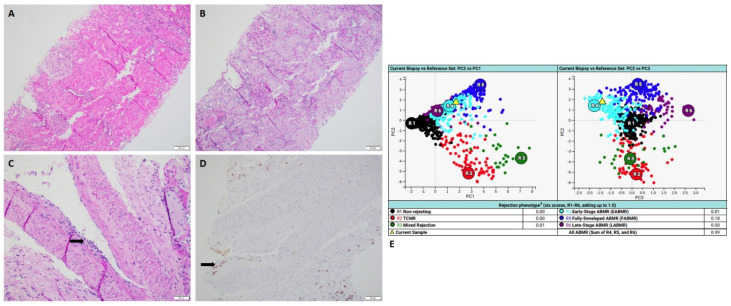
A case of isolated endarteritis with MMDx diagnosis of ABMR. (**A**,**B**) Hemoxylin and Eosin stains normal-appearing renal cortex with no inflammation (scale bar is 100 µm). (**C**) PAS stain showed isolated endarteritis with mononuclear cells infiltrating into the intimal layer of one artery (scale bar is 50 µm). (**D**) The immunohistochemical staining of CD3 highlighted the positive T-cell in the intimal layer of the artery (scale bar is 50 µm). (**E**) MMDx showed moderate early stage ABMR. (MMDx: molecular microscopy diagnostic system; ABMR: antibody-mediated rejection).

**Table 1 ijms-24-13817-t001:** Summary of discrepancies between histology and molecular diagnosis in a cohort of 94 renal allograft biopsies.

Discrepancy	Histology	MMDx	Clinical Information
Clear (*n* = 4)	Active BK nephropathy	moderate TCMR	BK viremia
	Active BK nephropathy	moderate TCMR	BK viremia
	CMV nephropathy	moderate TCMR	CMV viremia
	Pyelonephritis	mild TCMR	Pyelonephritis
Borderline (*n* = 2)	Isolated endarteritis with negative C4d staining	ABMR	Negative DSA
	Mild peritubular capillaritis (ptc1) with negative C4d staining	ABMR	Positive DSA

## Data Availability

No software products, custom code, or algorithms were developed for this manuscript. All resources are available to anyone upon request.

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
