# Peer review of "Discrepancy Analysis between Histology and Molecular Diagnoses in Kidney Allograft Biopsies: A Single-Center Experience"

_ijms, 2023, doi:10.3390/ijms241813817_

Round 1

Reviewer 1 Report

In the paper "Discrepancy... Experience", Suo et al provided evidence on the use of histology instead of molecular diagnosis of kidney transplant rejection. The authors argued that molecular testing of the biopsy materials does not provide the discrimination in presence of infections in such patients. The article is written concisely, the results presented are compelling and of considerable interest to the community of renal researchers and practitioners. The conclusions are justified by the evidence presented and address the main question posed in the article. I do not have any major comments. The manuscript requires some minor edits to fix the issues of definitions of acronyms used. Minor comments are as follows: 1. Line 43, define TCMR. 2. Line 71 define BK 3. Provide a brief explanation on why the research data cannot be shared. 4. Provide ethical statement as required by the journal policies on human and animal studies. 5. Methods section, provide the name of the institution where the IRB was housed. Add the statement about written informed consent and the research was conducted according to the principle of Declaration of Helsinki (1997).

Author Response

Thank you very much for taking the time to review this manuscript. Please find the detailed responses below and the corresponding revisions highlighted changes in the re-submitted files. Please see the attachments. 

Reviewer 2 Report

Authors submitted an interesting article research focusing on  histology diagnosis,  essential for the monitoring and management of kidney transplant patients. Nowadays, the accuracy and reproducibility of histology have been criticized when compared with molecular microscopy diagnostic system (MMDx). The cohort study resented  included 95 renal allograft biopsies with both histology and molecular diagnoses. Discrepancies between histology  and molecular diagnosis were assessed for each biopsy; Authors discussed clinical and laboratory assessment for  that cases they clearly illustrated.  

The significance of related pattern of histology,  molecular  and  clinical  aspects, with critical discussion by Authors,  may be useful for many other studies in this field.   

The study highlights that MMDx diagnosis can produce false positive results in the setting of active infections, especially viral infections. The MMDx Study Group indicated  that the diagnosis of TCMR by MMDx in the context of BK nephropathy can be clinically  significant and that the TCMR may resolve without treatment when full immunosuppression is restored .

active pyelonephritis. 

Author Response

Thank you very much for taking the time to review this manuscript. We really appreciate your time and effort.